# The Role of Antisiphon Devices in the Prevention of Central Ventricular Catheter Obliteration for Hydrocephalus: A 15-Years Institution’s Experience Retrospective Analysis

**DOI:** 10.3390/children9040493

**Published:** 2022-04-01

**Authors:** Dimitrios Panagopoulos, Georgios Strantzalis, Maro Gavra, Efstathios Boviatsis, Stefanos Korfias

**Affiliations:** 1Neurosurgical Department, Pediatric Hospital of Athens, 45701 Athens, Greece; 21st University Neurosurgical Department, Medical School, University of Athens, Evangelismos University Hospital, Athens 10676, Greece; ekne@neurosurgery.org.gr (G.S.); skorfias@med.uoa.gr (S.K.); 3Radiology Department, Pediatric Hospital of Athens, 45701 Athens, Greece; mmgavra@yahoo.com; 42nd University Neurosurgical Department, Medical School, University of Athens, Attikon General University Hospital, Athens 12462, Greece; eboviatsis@gmail.com

**Keywords:** shunt over-drainage, anti-siphon device, ventricular catheter obstruction, shunt revision

## Abstract

Shunt over-drainage in patients harboring a ventriculoperitoneal shunt constitutes one of the most devastating, and difficult to manage, side effects associated with this operation. Siphoning is one of the most important contributing factors that predispose to this complication. Based on the fact that the predisposing pathophysiologic mechanism is considerably multiplicated, amelioration of that adverse condition is considerably difficult to achieve. A lot of evidence suggests that the widespread utilization of gravitational valves or antisiphon devices is of utmost importance, in order to minimize or even avoid the occurrence of such complications. The recent literature data highlight that gravity-related, long-lasting shunt over-drainage consists of a momentous factor that could be considered one of the main culprits of central shunt failure. A lot of efforts have been performed, in order to design effective means that are aimed at annihilating siphoning. Our tenet was the investigation of the usefulness of the incorporation of an extra apparatus in the shunt system, capable of eliminating the impact of the siphoning effect, based on the experience that was gained by their long-term use in our institution. A retrospective analysis was performed, based on the data that were derived from our institution’s database, centered on patients to which an ASD was incorporated into their initial shunt device between 2006 and 2021. A combination of clinical, surgical, radiological findings, along with the relevant demographic characteristics of the patients were collected and analyzed. We attempted to compare the rates of shunt dysfunction, attributed to occlusion of the ventricular catheter, in a group of patients, before and after the incorporation of an anti-siphon device to all of them. A total number of 120 patients who have already been shunted due to hydrocephalus of different etiologies, were managed with the insertion of an ASD. These devices were inserted at different anatomical locations, which were located peripherally to the initially inserted valvular mechanism. The data that were collected from a subpopulation of 17 of these patients were subjected to a separate statistical analysis because they underwent a disproportionately large number of operations (i.e., >10-lifetime shunt revisions). These patients were studied separately as their medical records were complicated. The analysis of our records revealed that the secondary implementation of an ASD resulted in a decrease of the 1-year and 5-year central catheter dysfunction rates in all of our patients when compared with the relevant obstruction rates at the same time points prior to ASD insertion. According to our data, and in concordance with a lot of current literature reports, an ASD may offer a significant reduction in the obstruction rates that is related to the ventricular catheter of the shunt. These data could only be considered preliminary and need to be confirmed with prospective studies. Nevertheless, this study could be considered capable of providing supportive evidence that chronic shunt over-drainage is a crucial factor in the pathophysiology of shunt malfunction. Apart from that, it could provide pilot data that could be reviewed in order to organize further clinical and laboratory studies, aiming toward the assessment of optimal shunt valve systems that, along with ASD, resist siphoning.

## 1. Introduction

The overall handling of patients suffering from pediatric hydrocephalus remains a challenge for neurosurgeons, although it consists of a daily duty for all pediatric neurosurgical departments. This is true, although significant innovations in medical and engineering technologies have been reported during the last decades. It is a common concept that CSF diversion via the aid of ventricular shunts constitutes the main management option for all patient populations that are not candidates for endoscopic procedures [1,2,3]. A lot of technical improvements and modifications have been introduced in order to offer a considerable evolution of John Holter’s original differential pressure device [4]. This refers to the designation of a wide spectrum of valve technologies, which include flow-regulated, as well as adjustable valves [5,6]. The main motivation for the construction of all these subtypes of valvular mechanisms was the increasing frequency of recognized complications that should be attributed to the utilization of the existing shunt devices. Despite the different valve designations, serious and different devastating complications associated with shunting procedures are encountered, which continue to afflict the quality of life of the affected individuals. One of the most important and difficult to handle refers to the entity of shunt obstruction, the frequency of which stably remains at an unacceptably high rate [7,8,9,10]. According to the most recent research data, central catheter occlusion constitutes the most commonly recognized substrate, implicated in any case of shunt malfunction [9,10]. A wide spectrum of underlying etiologies is joined under the entity of central catheter obstruction, which includes malposition of the ventricular catheter, occlusion by choroid plexus or debris, as well as inflammatory changes [11,12]. Nevertheless, recent evidence supports the concept that chronic shunt over-drainage acts as a suction force, capable of entrapping ependymal tissues into the catheter orifices. The ultimate result of that process is catheter occlusion [13].

There exist anecdotal reports, referring to individual cases of immoderate drainage of cerebrospinal fluid, which are temporally placed in the early 1930s [14,15,16,17,18,19,20,21]. Subsequently, a lot of literature reports exist, centered on the definition of the constellation of signs and symptoms that constitute the entity of acute shunt siphoning [18,22]. 

According to physics, siphoning is the phenomenon where fluid continuously flows through an inverted U-shaped tube connecting two containers positioned at different heights. The fluid is “sucked” from the compartment with higher potential energy, flowing upwards against gravity to the “crown” of the system, and finally into the lower compartment. The flow continues until the hydrostatic pressure reaches equilibrium. Siphoning, or “immoderate drainage of CSF that is contained within the central nervous system cisterns” [23], is provoked whenever the affected individual adopts the upright posture, whereas they were lying supine. The net results of that posture alteration are the establishment of a pressure gradient, intimately related to gravity, which could potentially lead to a “negative hydrostatic suction force”. This could be considered as a precursor condition that ultimately leads to shunt over-drainage, and, secondarily, to ventricular collapse [24].

There exists considerable confusion regarding the exact rate of shunt over-drainage, mainly due to the lack of a universally accepted definition, which entails all the appropriate inclusion criteria. This fact is in accordance with the wide divergence that exists in the definition of the referred rate of this entity, which ranges from 1% to over 50% of shunted subjects [25]. In a recent survey that took place among ASPN members and was referring to patients suffering from hydrocephalus who underwent a ventriculoperitoneal shunt operation, most participants mentioned that, according to their clinical records, long-lasting excessive CSF drainage should be regarded as an uncommon complication of shunting (not exceeding 15%). They concluded that the reported chronic headaches should be attributed to other pathologic conditions (migraines, tension). Other studies support that the possibility of non-recognized (overlooked) shunt over-drainage should not be excluded in a subgroup of patients suffering from repeated shunt malfunctions [26], as over-drainage is a well-recognized predisposing factor, leading to occlusion of the central catheter [10]. As Rekate has mentioned, a minimum of one-third of individuals harboring long-standing shunts that are under clinical attention for more than a quinquennium will suffer from disabling long-lasting headaches. This may be in contradistinction to the fact that their clinical course could be devoid of complications in the long-term [27].

At this point, we would like to state that the premise that over-drainage is the main reason for proximal malfunctions is not firmly established in the literature. This means that describing over shunting as the “leading cause of shunt malfunction” is not currently a universally accepted theory.

Proposed treatment strategies for the treatment of excessive drainage of CSF are centered on two main targets. The first is to address symptom relief, and the other is centered on the confrontation of the etiology of over-drainage. Surgical interventions include replacement of the valvular mechanism, as well as the (additional) incorporation of ASDs [28]. Cumulative data support the concept that persistent shunt over-drainage should be considered as one of the most determinant incriminating factors in the context of repetitive proximal shunt occlusion [13]. As a continuation of that data, our team attempted to further augment the resistance to outflow of CSF implicated by a valvular mechanism, via the insertion of an ASD. Via the presentation of our study, we display our conclusions regarding the efficacy of ASDs, which are introduced in order to prevent or at least decrease the rate of proximal shunt obstruction. Moreover, as our report is consistent with previous publications [28,29], albeit its statistical significance is restricted by its retrospective design, it could be considered as a pilot study, whose data could be utilized in future prospective studies, aimed to suggest that addition of an ASD is associated with a remarkable decrease in the rate of central catheter occlusion. This statement is valid, regardless of the subgroup of patients that is under consideration (“simple” or “complex”).

## 2. Methods

### 2.1. Patient Characteristics

After giving consent by the Bioethics Committee of our Hospital (Pediatric Hospital of Athens, Athens, Greece), a retrospective analysis of institutional data was conducted, referring to pediatric participants (0–16 years old) suffering from hydrocephalus, irrespective of its underground pathophysiology. All of our participants share in common that an ASD was incorporated into their CSF drainage system, due to secondary complications, between 2006 and 2021. All relevant data were formally introduced in a prospectively collected surgical spreadsheet. Relevant patient information included primary cause of hydrocephalus, patient’s age at the time of first shunt implantation, specific technical characteristics of primary valvular mechanism, mode of clinical presentation that preceded surgical management, itemized description of all relevant surgical procedures that were executed prior to and following the insertion of ASD, as well as the location of the ASD.

Based on current literature data, the diagnosis of proximal shunt malfunction (as well as due to other causes, such as infection or distal catheter malfunction) was evidenced by the combination of evidence, derived from the clinical findings and brain imaging characteristics of any particular patient. When equivocal cases are encountered (i.e., when there is uncertainty regarding the underlying pathology, based on imaging studies), as well as in order to exclude the possibility of an underlying infection, a shunt tap was undertaken. The diagnosis of slit ventricle syndrome is established based on the combination of recurrent symptomatology associated with inappropriate shunt function, in combination with the small dimensions of the ventricular system. Shunt overdrainage with sudden onset is included in our differential diagnosis when symptomatology that could be attributed to positional changes is encountered, along with CT or MRI evidence compatible with a small-sized ventricular system. Intraoperatively, the central catheter could be regarded as being obliterated (non-functioning), in every case where one or both of the criteria that are mentioned below were encountered:Complete (or near complete) absence of flow through the central catheter, orThere is adhesion of the catheter to the surrounding brain, choroid plexus or debris, without any manifestation of spontaneous CSF flow.Absence of distal (to the ventricular catheter) flow of CSF, indicating valvular or distal catheter malfunction.

We would also like to mention that the Valsalva maneuver, gentle proximal flushing, and dropping of the head were performed, in order to confirm that the reason for no proximal flow was not from low pressure hydrocephalus.

Artificially, a separate group of patients was dissociated, incorporating records from individuals who had at least 10 shunt revisions since their initial shunt insertion. They were included under the term “complex” and considered separately from the “simple” subgroup of patients, which included all individuals who were submitted to less than 10 revisions each. This distinction (more than 10 versus less than 10 revisions) was selected from the beginning of our survey and it is a subjective dividing element. An identical criterion was used in a previous survey [1] and we agree with the estimation of the previously mentioned senior author that if the total amount of lifetime shunt reoperations is equivalent to or exceeds the number of 10, this could be regarded as a criterion that could fair enough differentiate a patient from the average failure rate. The majority of ASDs were inserted in individuals who were proved to be dependent on a well-functioning shunt system, and later on, they were complicated from the existence of sudden onset, or long-standing, CSF overdrainage. This subcategory of individuals shared in common several features, including repetitive central catheter occlusions and headaches that are related to the patient’s posture, or they are intractable, non-responding to any kind of medication. 

### 2.2. ASD Surgery

ASDs were inserted in line with the primary valvular mechanism, at a point peripheral to the position of the initial valvular mechanism, in close anatomical relationship with that and with its long axis preferably perpendicular to the floor. They were inserted mainly behind the ear, or, rarer at the neck, clavicle, chest, or abdomen levels, their relevant location being predicted in the majority of cases by previously performed skin incisions. Another important technical point was that every effort was executed in order to ensure that the long axis of the ASD was perpendicular to the floor (parallel to the longitudinal axis of the body), in order to maximize its efficacy to counteract the siphoning effect. Mini-NAV valves were used exclusively in premature neonates suffering from hemorrhagic hydrocephalus and in very young children, irrespective of the underlying pathophysiology of their hydrocephalus, due to their limited overall size (low valvular profile). 

### 2.3. Shunt Survival

Another tenet of our survey was to calculate the mean value of shunt survival rate that is relevant to the patient population under consideration. In order to collect all relevant data, we analyzed the medical data of all individuals that were treated in our hospital and were operated on (without additional ASD) in our department for insertion of ventriculoperitoneal shunt in the time period between 2003–2021. The endpoints were shunt revision (due to infection, obstruction, or even breakage of any component of the drainage system), and the overall shunt malfunction rate was calculated and juxtaposed to the relevant rates of shunt repair, as they are mentioned in the published series.

### 2.4. Statistical Analysis

Qualitative variables were expressed as absolute and relative frequencies. For the comparison of proportions, chi-square tests were used. All reported *p* values are two-tailed. Statistical significance was set at *p* < 0.05 and analyses were conducted using STATA statistical software (version 11.0). Only *p* values < 0.05 were considered to represent a statistically significant difference among the compared values (number of patients that were substituted to central catheter replacement due to obstruction).

## 3. Results

### 3.1. Epidemiology

#### Demographics and Hydrocephalus Etiology

A group of 120 individuals (0–16 years old) was submitted to an operation which included the insertion of an ASD at our institution, as a secondary operation, at the periphery of the initial shunt valvular mechanism, between 2006 and 2021. This population of patients was subdivided into male and female groups. Among them, 68 were male (57.27%) and 52 were female (42.73%). When the most common underlying pathophysiologic mechanisms that were involved in the pathogenesis of hydrocephalus were considered, several subcategories were identified. Post-hemorrhagic hydrocephalus which was attributed to intraventricular hemorrhage due to immaturity was the most common underlying mechanism (29%). Apart from that, other less frequently encountered pathological conditions included the existence of infra/supratentorial neoplastic lesions (25.45%), as well as myelomeningocele (19%) (Table 1).

### 3.2. Primary Valve

According to the data that refer to our cohort of ASD patients, the average age of the affected individuals when the initial shunt placement is considered, was 2 ½ years (median 0.5-years, range 1 day–16 years). The selected primary type of valvular mechanism (when the ASD was inserted) was a medium-pressure differential pressure valve (fixed opening pressure), and it was our choice in 33 patients (30%), followed by a low medium-pressure differential pressure valve (fixed opening pressure) (12.7%), and, finally a high -pressure differential pressure valve (fixed opening pressure) (6.3%). (Table 2).

### 3.3. ASD

The median age of the patients when they were considered for ASD placement was 6 years (average 9 years, range, 6 months–16 years old). The median time interval between the initial shunt placement and the reoperation, regarding the incorporation of the ASD, was 4 ½ years (average, 3-years, operated on range: 5 months–7 years). Out of the total number of patients, 65 (59.09%), were applied an ASD due to recurrent ventricular catheter obstructions. On the other hand, in the other 55 patients (40.9%), the underlying cause was intractable over-drainage that was related to devastating symptoms. A total number of 69 patients (62.7%) underwent simultaneous revision of the central (ventricular) catheter due to obstruction (malfunction), along with ASD placement (at the same time). Of these, 42 patients (60.86%) underwent replacement of the central catheter (insertion of another central catheter) because of central occlusion of various etiology, and another group of five patients necessitated simultaneous replacement of the valvular mechanism). An additional subgroup of 18 patients (26.08%) underwent revision of the distal (peripheral) catheter, whereas nine (13.04%) suffered from an infection of the drainage system, necessitating the incorporation of another one. The ASDs were intervened at the level of the retroauricular region (63.5%), clavicle (5.5%) and chest (31%). 

There is a separate group consisting of 17 patients, which shared in common the fact that they have undergone a minimum of 10 revisions of the shunt system, prior to adding an ASD to their drainage system. Albeit this separation criterion (>10 revisions) has been arbitrarily chosen, these patients are characterized by the peculiar complexity of their clinical course, so the data that were extracted from these patients were subject to separate statistical analysis and are described under the term “complex” individuals. This separation criterion was selected by another similar study in the recent past (super sos), which also attempted to clarify the exact role of ASDs. 

### 3.4. Baseline Shunt Failure Rate

A major limitation of our highly selective cohort of patients is based on the fact that it is not safe and scientifically correct to calculate the real shunt revision rates based on that data. In order to overcome this pitfall, we calculated the survival rates of the shunts that were inserted at our hospital, after their primary placement, based on our institution’s entire population of patients harboring a CSF drainage device without ASD, between 2000 and 2018. The relevant points of interest included shunt revision and infection. In this population of 298 patients, the percentage of functioning shunts (shunt survival rate) was 69.5% at 1-year, which diminished to 61.2% at the 2-year interval. This decline in the percentage of functioning shunts seems to be in accordance with other published series [1] (Table 3).

### 3.5. ASD Implantation and the Rate of Proximal Shunt Malfunction

The basic tenet of the current study was to exemplify the role, if any, of the ASDs in the reduction of the rate of proximal shunt failure, that is central catheter obstruction. In order to determine that, we reviewed the data regarding the shunt reoperation rates 1- and 5-years, prior to and following ASD placement respectively. As already mentioned, the participants were subdivided into those who had less than 10 shunt revisions during our observation period (“simple” patients), and, on the other hand, those with 10 or more shunt revisions (arbitrarily considered as “complex” patients).

### 3.6. “Simple” Patients (<10 Shunt Revisions)

This group numbers 93 patients, who underwent a total of 130 revisions of the shunt system in the time interval of 1-year before the insertion of the ASD. Among them, in a subgroup of 55 patients, 104 cases were recorded that involved occlusion of the central catheter of the shunt system. In the remaining 38 patients of this group, no operation was recorded, aiming toward revision of the proximal part of the shunt system, due to malfunction. Additionally, a sum of 17 cases was recorded, which was referring to a subgroup of 14 patients, which was related to malfunctioning of the peripheral catheter, and another group of nine cases (referring to seven patients), who were diagnosed and treated for a shunt infection.

In the time period of 1-year that followed the insertion of an ASD, a total of 52 revisions were performed in a population of 43 patients. More precisely, the number of patients who came to clinical attention due to obstruction of the ventricular catheter was reduced from 55 (who underwent 104 operations in total) to 18 patients (who underwent a sum of 21 revisions). This is equivalent to a reduction in the number of reoperations in the range of 67.2% (*p* < 0.001). There is a remaining subgroup of 25 patients, a total number of 31 revisions were recorded. In more detail, 17 patients underwent a total number of 20 reoperations due to dysfunction of the peritoneal catheter, whereas another group of eight patients underwent a total of 11 revisions, due to infectious complications that involved any part of the shunt system. The number of patients that did not come to clinical attention because of dysfunction that was attributed to the proximal catheter was increased from 38 patients to 75, accordingly (*p* < 0.001 Figure 1a,b). When the other two subcategories of revision are encountered, namely dysfunction of the peritoneal catheter and infection, there was an inability to establish a statistically significant difference when the data that refer to the time period before and after ASD insertion were analyzed.

Regarding the subgroup of the 93 ‘simple’ patients, our study incorporates recorded data for a 5-year time period (follow-up) before and after the administration of ASD. Regarding this subgroup of patients, a total of 284 revisions was encountered during the 5-year follow-up period that was preceding the incorporation of an ASD. A total of 67 patients was involved, suffering from proximal ventricular catheter obstruction, and 258 operations were executed, centered on the revision of the central catheter of the shunt system. On the contrary, there was a group consisting of three patients that were not submitted to any operation regarding the proximal shunt catheter. Moreover, a total number of 10 revisions were recorded, due to inappropriate function of the peritoneal catheter; these complications were attributed to a total number of nine patients. Besides that, our data recorded 14 cases of revision that were related to a shunt infection, concerning a total number of 14 patients. 

In the time interval that refers to the time period of 5-years after the incorporation of an ASD, our records verify that a total number of 102 revisions have been performed, which concern a total number of 78 patients. There was a reduction in the total number of patients who underwent revision due to malfunction of the central ventricular catheter. More precisely, in the corresponding time period prior to the insertion of an ASD, 67 patients underwent a total number of 258 revisions due to central catheter obstruction. In the 5-year period of follow-up that was initiated after the insertion of the ASD, 65 revisions were performed, concerning a subgroup of 30 patients. According to our statistical analysis, this is equivalent to a reduction in the order of 55.2% (*p* < 0.001). On the contrary, the total number of patients that did not suffer from central catheter obstruction was significantly increased. In more detail, we recorded an increase from 26 patients (from a total number of 93 patients included), to a number of 63 patients (when we compared the data of the 5-year follow-up period, before and after the insertion of an ASD, accordingly) (*p* < 0.001) Figure 1c,d). Additionally, 21 cases were recorded, suffering from a malfunction of the peripheral catheter, concerning a group of 18 patients, as well as 16 cases that appeared with shunt infection, attributable to 15 patients. When the relevant data of the two time periods (5-year before and after the ASD placement) were submitted to statistical processing, no significant difference was composed, regarding the incidence of shunt infection and peripheral catheter dysfunction. A total number of four cases were reported that concerned peripheral malfunction and this was due to disjunction (disconnection) at the level of ASD. As far as the cases of infection are concerned, three cases were located at the level of the incision that was utilized in order to connect the ASD. 

### 3.7. “Complex” Patients (10 or More Lifetime Shunt Revisions)

This group of patients shared in common the fact that the ASD was placed due to a history of repeated proximal obstruction. Our data revealed that during a time interval (follow-up period) of 1-year prior to placement of the ASD, 82 operations were performed, in a population of 17 patients. Among them, a total number of 56 cases were related to central catheter obstruction and these cases referred to a group of 13 patients. On the contrary, there was another group, consisting of four patients, in whom no revision was performed due to central catheter obstruction. Additionally, during this time period, we managed 20 cases that were referred to us due to peripheral catheter obstruction, in a subgroup of 10 patients, as well as six cases with an infection that was attributed to a total number of five patients. 

Moreover, data were collected for the time period of 1-year after the placement of the ASD. During this period, a total of 23 revisions were performed, on a number of 10 patients. When that data was compared with the aforementioned (1-year before the administration of the ASD), there was a reduction in the number of patients that underwent revision of the proximal ventricular catheter (a total of five patients). This subset of patients underwent a total number of 10 proximal catheter revisions (reduction 61.5%, *p* = 0.006). Moreover, another subgroup of 10 patients underwent a total of 18 operations, due to malfunction of the peritoneal catheter, and another subset of five patients was operated on due to shunt-related infection (total number of six re-operations). The aforementioned data (regarding the peritoneal catheter and the shunt related infection cases) does not display any statistically significant correlation with the addition of the ASD. Moreover, no statistically significant difference was recorded in conjunction with the data that were referring to the time period prior to the ASD administration. When the number of patients that did not reveal clinical evidence of ventricular catheter obstruction was concerned, we recorded an increase in their total number from four to 12 patients (*p* = 0.006 Figure 2a,b).

Regarding the same subgroup (‘complex’ patients), our survey has recorded clinical and surgical data for a time period of 5-years before and after the administration of an ASD. During this time period prior to the utilization of an ASD, a total number of 218 revisions were executed in this subgroup of 17 patients. In a subgroup of 16 patients, a total number of 172 cases were recorded that were referring to central (ventricular) catheter obstruction only one patient was not involved with central catheter malfunction. On the other hand, a subpopulation of 12 patients was recorded because they presented with peripheral catheter dysfunction (a total number of 35 revisions). Finally, 11 cases that were complicated with shunt infection were recorded, concerning a total number of seven patients.

In the ‘complex’ subgroup of patients, data were recorded for a time interval of 5-years after the insertion of the ASD. During this follow-up period, a total number of 12 patients underwent 37 revisions in total. The total number of patients that were implicated with a malfunction of the central catheter was reduced from 16 (who underwent a total of 172 operations) to six patients (who underwent 25 operations in total) (reduction 62.5%, *p* < 0.001). Another important notion is that the total number of patients that did not manifest with central catheter obstruction was increased from one to 11 patients (*p* < 0.001 Figure 2c,d).

Our data also recorded a total number of five patients that were implicated with dysfunction of the peritoneal catheter (total, six operations recorded) and also another subgroup of five patients who demonstrated shunt infection (a total number of six operations). These data (regarding the dysfunction of the peritoneal catheter and the infection) did not reveal any statistically significant difference when they were compared with their counterparts, regarding the same time interval prior to the administration of the ASD. Apart from that, there is a complete lack of evidence that any one of these complications is related to the usage of ASD. Finally, we mention that at the time of the initial shunt placement, we preferred the insertion of a ventriculoperitoneal shunt placement in a subgroup of 110, out of 120 patients. In the remaining subgroup of 10 patients, we preferred the option of inserting a ventriculoatrial shunt (via the internal jugular vein). During the follow-up period, in a small subgroup of six patients, a revision of the ventriculoperitoneal shunt system was performed, due to a malfunction of the peripheral catheter. In these cases, the ventriculoperitoneal shunt system was removed, and instead of that, a ventriculoatrial shunt was selected as a salvage procedure. This means that at the end of our survey, a ventriculoperitoneal shunt was selected in 104 patients, which is 85.45% of the total number of participants. On the other hand, a ventriculoatrial shunt was introduced in the remaining 16 patients (14.54%). This fact was not able to differentiate the degree of statistical significance regarding the results of our survey.

Table 4, Table 5, Table 6 and Table 7 compare all the described variables between non-ASD and ASD groups.

## 4. Discussion

Based on our statistical analysis, as well as on our cumulative clinical experience, it seems that ASD insertion is inherently related to a considerable decrease, in the relevant rates of central catheter malfunction due to obstruction. This fact could be attributed to the additional resistance that is exercised by the ASD against the flow of CSF, in addition to the one exerted by the conventional valvular mechanism. Regardless of the propensity of these patients to suffer from repetitive shunt malfunctions prior to the utilization of ASDs, there is enough evidence to support the fact that there is a reduced 1-year and 5-year post ASD obstruction rate, evident in the vast majority of patients. Our evidence is also in concordance with other data presented in the literature regarding the pre-ASD shunt malfunction rate (Table 3). Another important notification is centered on the fact that these effects are consistent, as there is evidence that supports the durability of this effect 5-years after ASD insertion. This fact is further enhanced by the remark that this effect is notable, and statistically significant, in the subgroup of “complex” patients. This group of patients was studied separately from the other patient population, due to its predisposition to being subjected to a disproportionately larger figure of shunt malfunctions. An underlying mechanism that is gravity-dependent is proposed in the literature as the offending substrate for these cases of proximal catheter obstruction [13]. Our results enhance the concept that gravity-driven shunt over-drainage is the main underlying pathophysiology associated with proximal shunt obstruction and enhance the validity of previous reports which highlighted the use of ASDs as an effective means in our effort to ameliorate proximal shunt obstruction rates [28,29].

### 4.1. Shunts and ASDs

The concept of drainage of excess CSF via a shunt device remains the most effective and widely used alternative in our therapeutic armamentarium regarding the management of hydrocephalus, irrespective of its underlying pathophysiology. This is currently the case, irrespective of the high failure rates associated with shunt insertion [9], along with their relevant long-standing adverse effects [7,8,9,30]. The incorporation of an ASD into a shunt system, aiming to decrease over-drainage, has been a well-established practice in the surgical treatment algorithm of these patients, for several decades [23,28,29,31,32]. The first reported attempt is placed chronologically in the early 1970s when Portnoy et al. investigated the first version of an ASD [32].

The ultimate goal of the prototype ASD was to eliminate the siphoning effect that happens when patients that harbor a CSF shunt adopt a standing posture, due to the pressure slopes that are developed between the cranial and peritoneal cavity. Although a lot of years have passed since their initial utilization, their use is not devoid of limitations and controversies [24]. While researchers have adopted ASD implantation to manage patients suffering from “chronic shunt overdrainage” [33,34], as well as to achieve re-expansion of slit ventricles [35,36] and reduction of shunt malfunction rates [28,29], another group of scientists has cautioned about their potential drawbacks. The addition of ASD has been correlated with an increased risk of underdrainage, accompanied by symptomatic ventriculomegaly, especially in chronically bed-ridden patients [37,38].

At this point, we would like to mention that the use of a Kaplan–Meier survival curve could be proposed as an efficient means to potentially improve the presentation of our data. We would like to state that two similar studies, focused on the relevant efficacy of anti-siphon devices and programmable valves, respectively [1,39], in the management of shunt over-drainage, both performed a statistical analysis that shared a lot with our study.

Additionally, it would be very interesting to also see the difference between the antigravitational system and the installation of a valve in high pressure to observe if we observe the same effect. To the best of our knowledge, there is one relevant, recent report [39] that is centered on the usefulness of programmable valves, to counteract the over-drainage phenomenon. They concluded that an upgrade of a programmable valve is an adequate means to handle the consequences of shunt over-drainage. This manipulation is very similar to a suggestion, regarding the installation of a valve in high pressure, to observe if we observe the same effect. Although there is no current literature that could answer the inquiry, our scientific positioning is that it could offer similar results. A large, prospective cohort study should be organized, comparing two groups of patients, suffering from over-drainage, who share in common all other demographic and underlying pathophysiologic characteristics. Patients that belong to one group should be managed by up-regulating their valvular mechanism, whereas the other group of patients should be offered the possibility to insert an anti-siphon device.

### 4.2. Limitations and Further Issues

The inherent restriction that is associated with this study is mainly related to its endogenous characteristics. This mainly consists of the fact that it is based on the data derived from a single-center, the number of participants is relatively small, and it is a retrospective analysis, whose reference base is composed of a heterogeneous population of patients. Apart from that, and after taking into consideration the fact that we are unable to exclude the possibility that a selection bias has occurred, this factor is unable to completely abolish the statistical significance of our results. We recognize that inherent limitations are consistent with a study that is planned in a retrospective model. Nevertheless, this fact is insufficient to completely cancel the significance of our results. More precisely, we cannot ignore the fact that a significant proportion of the shunt population responded positively and in the long-term, to our management protocol. It seems that our data are in concordance with the concept that central catheter occlusion is strongly associated with the existence of chronic shunt over-drainage, even though they should be validated with clinical trials or registries. Moreover, there are a lot of issues that need further investigation, including the determination (if it exists) of a specific ASD mechanism that is more efficient in eliminating the risk of shunt dysfunction, and where is the preferred site of insertion regarding the ASD, taking always into consideration the position of the primary valve. Finally, the investigation of any possible association that exists between chronic shunt drainage and the secondary development of pathological brain compliance and ASD function, is an issue that deserves special mention.

## 5. Conclusions

The conclusions that are extracted, based on our institution’s clinical records are in line with the data that are extracted from other surveys. More precisely, these data support the fact that the addition of a mechanism, in line with the original valvular mechanism, able to decrease the rate of CSF outflow, namely the ASD, is capable of achieving a remarkable reduction in the rate of occlusion of the central catheter of the shunt system. We recognize that our study has inherent limitations. These are associated with its retrospective design, the fact that it is based on a patient population that consists of a heterogeneous group of participants with dissimilar underlying pathophysiologic substrates regarding the etiology of the hydrocephalus, the existence of a wide spectrum regarding the age of the affected individual when the initial shunt insertion was performed, and the wide variety that refers to the selected primary valvular mechanism and ASD subtype. Nevertheless, the evidence that is extracted from our study is in concordance with other surveys which underline the usefulness of prospective trials to elucidate the usefulness of ASDs in the avoidance of CSF over-drainage in shunted patients. If our data are further confirmed, this survey would provide valuable support to the concept that long-standing shunt over-drainage is a significant parameter that could be associated with suboptimal shunt function. Our basic tenet is to provide pilot data that would be able to further guide clinical and laboratory studies to better determine the optimum ASD type and its preferred site of insertion. Our ultimate goal would be the enhancement of our effort to develop an integrated shunt-valvular system that is altogether capable to provide resistance to siphoning.

## Figures and Tables

**Figure 1 children-09-00493-f001:**
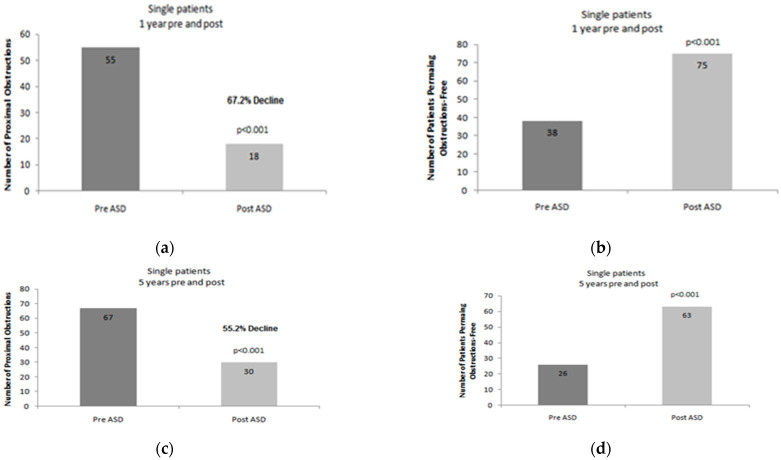
Proximal shunt revision rates after ASD placement in “simple” patients. (**a**): Graphical illustration of the 1-year number of proximal shunt revisions after ASD implantation in “simple” shunt patients. (**b**): Graphical illustration of the number of patients with no proximal shunt obstruction during the study period (1-year). (**c**): Graphical illustration of the 5-year number of proximal shunt revisions after ASD implantation in “simple” shunt patients. (**d**): Graphical illustration of the number of patients with no proximal shunt obstruction during the study period (5-years).

**Figure 2 children-09-00493-f002:**
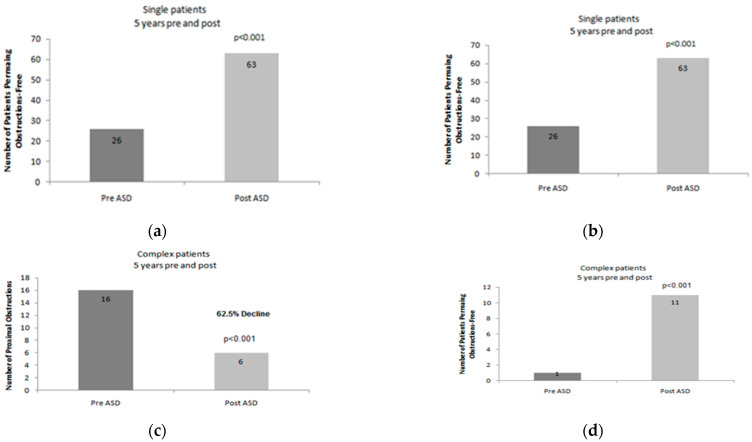
Proximal shunt revision rates after ASD placement in “complex” patients. (**a**): Graphical illustration of the 1-year number of proximal shunt revisions after ASD implantation in “complex” shunt patients. (**b**): Graphical illustration of the number of patients with no proximal shunt obstruction during the study period (1-year). (**c**): Graphical illustration of the 5-year number of proximal shunt revisions after ASD implantation in “complex” shunt patients. (**d**): Graphical illustration of the number of patients with no proximal shunt obstruction during the study period (5-years).

**Table 1 children-09-00493-t001:** Patient epidemiology.

Variable	No. of Patients (%)
Gender	
Male	68
Female	52
Etiology of hydrocephalus	
Infantile posthemorrhagic hydrocephalus	45
Neoplasm (Supra/infra-tentorial)	33
Myelomeningocele	11
Congenital Hydrocephalus (i.e., aqueductal stenosis)	14
Arachnoid Cysts	5
Post-meningitis	3
Posttraumatic	3
Idiopathic Intracranial Hypertension	2
Chiari Malformation	2
Dandy-Walker syndrome	2
Patient age at primary shunt placement	
<1-year	65
<1 month	31
1–6 months	23
6–12 months	11
1–10 months	41
>10-years	14
Patient age at ASD placement (not corrected for immaturity) (years)	
<1-year	17
1–5-years	23
5–10-years	39
10–16 years	41

**Table 2 children-09-00493-t002:** Types of primary valvular mechanisms and ASDs.

Primary Valvular Mechanism	No. of Patients (%)
Low opening pressure	14
Medium opening pressure	36
High opening pressure	9
Strata	5
Delta	4
Codman	5
OSV	2
Accessory ASDs	6
PaediGav 9/19	10
Shunt Assistant 00/15	7
Shunt Assistant 00/20	5
Lower-profile accessory ASDs	5

**Table 3 children-09-00493-t003:** Summary of 1- and 2-year shunt failure rates in different published shunt studies.

Series	Region	No of Pts	% Shunt Survival after 1-Year	% Shunt Survival after 2-Years
Liptak & McDonald, 1985–1986	USA	149	59	50
Sainte-Rose et al., 1991–1992	Canada	1620	71	-
Drake et al., 1998	Canada	344	61	47
Zemack et al., 2003	Sweden	158	60.5	53
Hanlo et al., 2003	Germany	557	71	67
Shannon et al., 2012	USA	338	-	51
Al-Hakim et al., 2018	Germany	116	68	-
Koueik et al., 2019	USA	168	70.23	64.28
Current Study	Greece	298	69.5	61.2

**Table 4 children-09-00493-t004:** Comparison of the number of patients who underwent revision of the proximal catheter in a time period of one year before and one year after the insertion of an ASD, in the subgroup of ‘Simple’ patients.

‘Simple’ Patients (Total Number: 93)	
1-year before the insertion of an ASD	
No of patients	55
Total number of central catheter revisions	104
1-year after the insertion of an ASD	
No of patients	18
Total number of central catheter revisions	21

**Table 5 children-09-00493-t005:** Comparison of the number of patients who underwent revision of the proximal catheter in a time period of five years before and five years after the insertion of an ASD, in the subgroup of ‘Simple’ patients.

‘Simple’ Patients (Total Number: 93)	
5-years before the insertion of an ASD	
No of patients	67
Total number of central catheter revisions	258
5-years after the insertion of an ASD	
No of patients	30
Total number of central catheter revisions	65

**Table 6 children-09-00493-t006:** Comparison of the number of patients who underwent revision of the proximal catheter in a time period of one year before and one year after the insertion of an ASD, in the subgroup of ‘Complex’ patients.

‘Complex’ Patients (Total Number: 17)	
1-year before the insertion of an ASD	
No of patients	13
Total number of central catheter revisions	56
1-year after the insertion of an ASD	
No of patients	5
Total number of central catheter revisions	10

**Table 7 children-09-00493-t007:** Comparison of the number of patients who underwent revision of the proximal catheter in a time period of five years before and five years after the insertion of an ASD, in the subgroup of ‘Complex’ patients.

‘Complex’ Patients (Total Number: 17)	
5-years before the insertion of an ASD	
No of patients	16
Total number of central catheter revisions	172
5-years after the insertion of an ASD	
No of patients	6
Total number of central catheter revisions	25

## Data Availability

Data is contained within the article.

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
