# Peer review of "The Role of Antisiphon Devices in the Prevention of Central Ventricular Catheter Obliteration for Hydrocephalus: A 15-Years Institution’s Experience Retrospective Analysis"

_children, 2022, doi:10.3390/children9040493_

Round 1
Reviewer 1 Report
tthe number of patients is substantial thus highlighting your results.
the form is more difficult and it would be interesting to revise the introduction as well as the discussion in order to emphasize the subject of your study. the bibliographical references are good.
it would be very interesting to also see the difference between the antigravitational system and the installation of a valve in high pressure in order to observe if we observe the same effect.
this point needs to be better specified in the manuscript.
Author Response
Dear Reviewer 1,
Thank you for your valuable comments. We have carefully studied your remarks, and we would like to state the following, in an attempt to adequately cover your report.
You have mentioned that ‘the form is more difficult and it would be interesting to revise the introduction as well as the discussion in order to emphasize the subject of your study.’
We really appreciate your suggestion. All necessary adjustments are included in our revised section in order to comply with your remarks.
Additionally, you have mentioned that ‘it would be very interesting to also see the difference between the antigravitational system and the installation of a valve in high pressure in order to observe if we observe the same effect. this point needs to be better specified in the manuscript.’
We strongly consider that your enquiry is important to be adequately considered. To the best of our knowledge, there is one relevant, recent report that is centered on the usefulness of programmable valves, in order to counteract the over-drainage phenomenon. They concluded that an upgrade of a programmable valve is an adequate means in order to handle the consequences of shunt over-drainage. This manipulation is very similar with your suggestion, regarding the installation of a valve in high pressure, in order to observe if we observe the same effect. Although there is no current literature that could definitely answer your enquiry, our scientific positioning is that it could offer similar results. In order to definitely answer to your point, a large, prospective cohort study should be organized, comparing two groups of patients, suffering from over-drainage, who share in common all other demographic and underlying pathophysiologic characteristics. Patients that belong to one group should be managed by up-regulating of their valvular mechanism, whereas the other group of patients should offer the possibility to insert an anti-siphon device.
A relevant statement is added to our revised section of our manuscript.
Reviewer 2 Report
Panagopoulus et al. report report a retrospective, observational study examining the utility of anti-siphon devices in preventing proximal catheter obstruciton after VP shunt implantation. The theory put forward was this preventing overshunting induced adhesion of soft tissue to the catheter. 120 patients were included from 2006 to 2021 who at some time point had a ASD placed. Reduced rates of proximal catheter obstruction were found at the 1 and 5 year mark. The authors would be better served in their intent by focusing more on each revision as a new case and clearly illustrate the time until revision/indication for revision when a ASD vs no ASD Is present. AS it is currently written, it is very difficult to understand the comparison group/ number of non-ASD valves being compared. A table that compares all the described variables between non-asd and asd groups would be very helpful.
Was Valsalva, gentle proximal flushing, or dropping the head performed to confirm the reason for now proximal flow wasn’t from low pressure hydrocephalus?
Author Response
Dear Reviewer 2,
Thank you for your valuable comments. We have carefully studied your remarks, and we would like to state the following, in an attempt to adequately cover your report.
You have mentioned that ’The authors would be better served in their intent by focusing more on each revision as a new case and clearly illustrate the time until revision/indication for revision when a ASD vs no ASD Is present. AS it is currently written, it is very difficult to understand the comparison group/ number of non-ASD valves being compared. A table that compares all the described variables between non-asd and asd groups would be very helpful.’
We really appreciate your suggestion. All necessary adjustments are included in our revised section in order to comply with your remarks.
You have also written that ‘Was Valsalva, gentle proximal flushing, or dropping the head performed to confirm the reason for now proximal flow wasn’t from low pressure hydrocephalus?’
We apologize for our oversight. We state that all of the aforementioned maneuvers were undertaken, in order to rule out a low-pressure hydrocephalus. A relevant correction is added to our revised manuscript.
Table 4. Comparison of the number of patients who underwent revision of the proximal catheter in a time period of one year before and one year after the insertion of an ASD, in the subgroup of ‘Simple’ patients.
‘Simple’ patients (total number: 93) |
|
1 year before the insertion of an ASD |
|
No of patients |
55 |
Total number of central catheter revisions |
104 |
1 year after the insertion of an ASD |
|
No of patients |
18 |
Total number of central catheter revisions |
21 |
Table 5. Comparison of the number of patients who underwent revision of the proximal catheter in a time period of five years before and five years after the insertion of an ASD, in the subgroup of ‘Simple’ patients.
‘Simple’ patients (total number: 93) |
|
5 years before the insertion of an ASD |
|
No of patients |
67 |
Total number of central catheter revisions |
258 |
5 years after the insertion of an ASD |
|
No of patients |
30 |
Total number of central catheter revisions |
65 |
Table 6. Comparison of the number of patients who underwent revision of the proximal catheter in a time period of one year before and one year after the insertion of an ASD, in the subgroup of ‘Complex’ patients.
‘Complex’ patients (total number: 17) |
|
1 year before the insertion of an ASD |
|
No of patients |
13 |
Total number of central catheter revisions |
56 |
1 year after the insertion of an ASD |
|
No of patients |
05 |
Total number of central catheter revisions |
10 |
Table 7. Comparison of the number of patients who underwent revision of the proximal catheter in a time period of five years before and five years after the insertion of an ASD, in the subgroup of ‘Complex’ patients.
‘Complex’ patients (total number: 17) |
|
5 years before the insertion of an ASD |
|
No of patients |
16 |
Total number of central catheter revisions |
172 |
5 years after the insertion of an ASD |
|
No of patients |
06 |
Total number of central catheter revisions |
25 |
Reviewer 3 Report
First and foremost, the grammar, style, and language errors in this manuscript make it extremely difficult to determine the validity of the methods used and, therefore, the validity of the data presented.
One way to improve the presentation of data would be to use a Kaplan Meier survival curve rather than the manner in which the data is currently presented.
Finally, the authors should consider making their language more objective. For instance, the premise that over-drainage is the main reason for proximal malfunctions is not firmly established in the literature, and their references are appropriately speculative on this point. As such, describing over shunting as the "leading cause of shunt malfunction" detracts from what would otherwise be an interesting manuscript.
Author Response
Dear Reviewer 3,
Thank you for your valuable comments. We have carefully studied your remarks, and we would like to state the following, in an attempt to adequately cover your report.
You have reported that ‘First and foremost, the grammar, style, and language errors in this manuscript make it extremely difficult to determine the validity of the methods used and, therefore, the validity of the data presented. One way to improve the presentation of data would be to use a Kaplan Meier survival curve rather than the manner in which the data is currently presented.’
We really appreciate your valuable comments. We have performed a language editing in order to ameliorate the grammar, style, and language errors.
As far as the use of a Kaplan Meier survival curve, we would like to mention that two similar studies, focused on the relevant efficacy of anti-siphon devices and programmable valves, respectively, in the management of shunt over-drainage, they both performed a statistical analysis that shared a lot together with our study. We have modified accordingly our manuscript, adding a relevant statement.
You also mentioned that ‘Finally, the authors should consider making their language more objective. For instance, the premise that over-drainage is the main reason for proximal malfunctions is not firmly established in the literature, and their references are appropriately speculative on this point. As such, describing over shunting as the "leading cause of shunt malfunction" detracts from what would otherwise be an interesting manuscript.’
We really appreciate your valuable comments. We have modified accordingly our manuscript, adding a relevant statement.
Round 2
Reviewer 2 Report
I accept these changes as adequate.
Reviewer 3 Report
The authors have not addressed my concerns so I cannot endorse this manuscript.